# Feasibility and Efficacy of Telehealth-Based Resistance Exercise Training in Adolescents with Cystic Fibrosis and Glucose Intolerance

**DOI:** 10.3390/ijerph19063297

**Published:** 2022-03-11

**Authors:** Clifton J. Holmes, Susan B. Racette, Leslie Symonds, Ana Maria Arbeláez, Chao Cao, Andrea Granados

**Affiliations:** 1Program in Physical Therapy, Washington University School of Medicine, St. Louis, MO 63110, USA; racettes@wustl.edu (S.B.R.); caochao@wustl.edu (C.C.); 2Department of Medicine, Center for Human Nutrition, Washington University School of Medicine, St. Louis, MO 63110, USA; 3Department of Pediatrics, Washington University School of Medicine, St. Louis, MO 63110, USA; lsymonds@wustl.edu (L.S.); aarbelaez@wustl.edu (A.M.A.); 4Department of Pediatrics, Division of Endocrinology and Metabolism, Nicklaus Children’s Hospital, Miami, FL 33155, USA; andrea.granados@nicklaushealth.org

**Keywords:** cystic-fibrosis-related diabetes, fitness, body composition, telemedicine, COVID-19

## Abstract

The aims of this study were to (1) determine the feasibility of a home-based resistance exercise training (RET) program in patients with cystic fibrosis and impaired glucose tolerance using virtual personal training and (2) observe the effects completion of the RET program had on glucose metabolism, pulmonary function, body composition, and physical fitness. The feasibility of the program was defined as 80% compliance. Ten participants (15.80 ± 2.20 yr, 25.1 ± 7.4 kg/m^2^) began a home-based resistance training program consisting of 36 sessions supervised via online videoconferencing. Compliance scores of 78.9% (all participants) and 81.8% (without one outlier) were observed. A significant increase was observed in 2-h C-peptide levels (2.1 ng/mL; *p* = 0.04), with a moderate decrease in fasting glucose (−5.2 mg/dL; *p* = 0.11) and a moderate increase in 2-h insulin (35.0 U/mL; *p* = 0.10). A small decrease in the fat percentage (−1.3%; *p* = 0.03) was observed in addition to increases in fat-free mass (1.5 kg; *p* = 0.01) and the fat-free mass index (0.4; *p* = 0.01). Small, yet statistically significant increases were observed in V̇O_2peak_ (0.1 L/min *p* = 0.01), V̇CO_2peak_ (0.1 L/min; *p* = 0.01), and ventilation (5.3 L/min; *p* = 0.04). Telehealth-based RET is feasible in adolescents with CF and impaired glucose tolerance and elicits small yet favorable changes in insulin secretion, body composition, and exercise capacity.

## 1. Background

Cystic fibrosis (CF) is an inherited disease primarily affecting breathing and digestion, which results in shortened lifespan due to pulmonary failure and multi-systemic complications [1]. Cystic-fibrosis-related diabetes (CFRD) is the most common co-morbidity of CF, affecting at least half of the adult population, and is associated with a compromise of the nutritional status and body composition and a decline in pulmonary function. In the general population, strategies using exercise training play a role in glycemic control, mainly though improvements in insulin sensitivity [2,3,4]. Regular participation in physical activity and engagement in structured exercise training programs is strongly encouraged for individuals with CF, as they have been shown to improve exercise tolerance, pulmonary function, energy levels, and quality of life [5,6]. Resistance exercise training (RET) is an established method for increasing lean body mass and muscle strength. Studies in healthy individuals, both recreationally active and sedentary, have shown that RET can increase maximal oxygen consumption (V̇O_2max_), time to exhaustion, and maximal work rate during cycle ergometer testing [7,8,9]. Moreover, V̇O_2max_ correlates with mortality in children and adults with CF [10]. RET was also shown to reduce glycosylated hemoglobin and enhance glycemic control through increased glucose uptake and an enhancement of insulin action [11,12,13,14,15]. However, there is a lack of controlled trials evaluating the effects of a resistance training program on glucose dysregulation in adolescents and young adults with cystic fibrosis and glucose intolerance.

The coronavirus disease 2019 (COVID-19) global pandemic has dramatically affected how people carry out their daily lives, giving rise to physical distancing restrictions and specific protocols for home isolation or quarantining. Though COVID-19 poses a health risk to everyone, according to the Centers for Disease Control and Prevention, individuals with chronic respiratory diseases, such as CF, are at a higher risk for severe COVID-19 [16]. This presents unique challenges for clinicians, researchers, and other practitioners to conduct clinical studies traditionally conducted in-person in laboratory or hospital settings. Telehealth, or telemedicine, describes distance-based interventions performed using communication technologies (e.g., phone calls, text messages, and videoconferencing) to assess, educate, monitor, and/or deliver health-related programs, including exercise [17]. Though the emergence of telehealth has grown in prominence, its efficacy to deliver quality and effective results compared to traditional means is still unclear. Therefore, the aims of this pilot study were to (1) determine the feasibility of a home-based resistance exercise training program in patients with CF and abnormal glucose tolerance using virtual personal training and (2) observe the effects completion of the RET program had on glucose metabolism, pulmonary function, body composition, and physical fitness. We hypothesized that a virtually supervised RET would be feasible and well tolerated by adolescents with CF and abnormal glucose tolerance. Moreover, we hypothesized that RET would improve fasting and 2-h glucose measures during oral glucose tolerance testing (OGTT), enhance pulmonary function, and increase exercise tolerance and strength.

## 2. Materials and Methods

### 2.1. Participants and Study Design

Adolescents aged 10 to 18 years with CF and pancreatic insufficiency were invited to participate in this prospective study. Additional eligibility criteria were that participants had to be clinically stable, without evidence of deterioration from previous pulmonary function tests, and no hospitalizations or steroid use for 1 month prior to the study visit. Participants were included in the study if they had a history of CFRD without fasting hyperglycemia and were not on insulin therapy. Eligible diagnoses included impaired fasting glucose (≥100–125 mg/dL), impaired glucose tolerance (2-h glucose 140–199 mg/dL), or indeterminate glycemia (1-h glucose ≥ 200 mg/dL and normal fasting and 2-h glucose). Exclusion criteria included the use of medications affecting glucose homeostasis (insulin included), pulmonary exacerbations or hospital admissions in the 4 weeks prior to the study visit, and pregnancy. All participants and their parents or legal guardians signed informed consent or provided verbal assent (if <18 yr). This study was approved by the Washington University in St. Louis Institutional Review Board (IRB#201806163). Participants attended the Washington University CF Center for clinical visits and the Washington University Clinical Translation Research Unit for research study visits. Participants completed two study assessment visits, one at baseline and the other after a home-based resistance exercise training intervention. Participants fasted for a minimum of 8 h (ad libitum water intake was allowed) before both study visits and were instructed to refrain from exercising for at least 24 h prior to testing.

### 2.2. Body Composition Assessments

Criterion whole body measures of body composition were obtained using a Lunar Prodigy Advance DXA scanner (General Electric Healthcare, Encore Software Version 16). Prior to each scan, the DXA was calibrated according to the manufacturer’s instructions using a standard calibration block. Whole body scans were performed with participants in a supine position on the scanning bed with hands at their side. The following whole-body derivative values were calculated using the following equations:Fat Mass (FM) = Body Mass × (%Fat/100)
Fat-Free Mass (FFM) = Body Mass − FM
Fat Mass Index (FMI): FM/height^2^
Fat-Free Mass Index (FFMI): FFM/height^2^
Lean Body Mass Index (LBMI): lean mass/height^2^).

LBMI and FMI were converted to sex- and age-adjusted *z*-scores for the participants in the study.

### 2.3. Physical Fitness Assessments

To determine muscular strength, participants completed maximal isometric and isokinetic knee extension testing on a Biodex System 4 Isokinetic Dynamometer (Shirley, NY, USA). Isometric testing on the dominant leg was repeated three times where participants contracted maximally for five seconds with two minutes of rest between contractions. The average of the two highest peak torque values and the corresponding time to peak torque values were used in the final analysis. Isokinetic peak torque, work, average power, and time to peak torque were measured at 90°/s 180°/s, 270°/s, and 360°/s. The average of three trials at each speed was calculated for the final analysis.

Following the strength assessment, participants performed a maximal graded exercise test (GXT) using a ramped protocol on a recumbent cycle ergometer (Lode, The Netherlands) to determine peak oxygen consumption (V̇O_2peak_). Continuous heart rate and oxygen uptake measurements were collected using a 12-lead electrocardiograph set up and open-circuit spirometry with the Parvomedics Truemax 2400 computerized metabolic system (Parvomedics, Salt Lake City, UT, USA), respectively. After a brief warm-up, participants cycled at a pedaling rate 60 revolutions/min with resistance starting at 10 W/min that was increased each minute by 10–20 W/min until volitional exhaustion. The rating of perceived exertion (RPE) was measured during the final 30 s of each exercise stage using the Borg 6–20 scale [18]. During testing, participants cycled continuously while breathing through a special mouthpiece to measure oxygen uptake. The oxygen uptake values were calculated as the highest 15-s average achieved during the GXT. Exercise testing was considered a “peak” performance if there was a plateau in oxygen consumption despite an increased work (±2 mL/kg/min^−1^), respiratory exchange ratio of ≥1.10, and/or a heart rate (HR) was ≥85% of the age-predicted maximum (220-age).

### 2.4. Resistance Exercise Training Intervention

After baseline testing, participants began a home-based resistance training program consisting of 36 whole-body exercise sessions performed 3 times per week on nonconsecutive days, allowing for at least 24 h of recovery between sessions. Completion of the 36 sessions was anticipated within a 12-week period; however, an additional 2 weeks were given to account for missed sessions due to unforeseen situations (e.g., sickness or travel). The participants were given a set of weight-adjustable dumbbells and assigned a personal trainer to supervise their at-home, one-on-one sessions via live video calls using a web based platform (i.e., Zoom). The trainers would provide instruction, demonstration, and verbal encouragement during all sessions with each participant. Additionally, a minimum of one adult parent/guardian was tasked to be present in person at the participant’s home during all training sessions in case of injury or other emergency. Each personal trainer kept a detailed log of completed sessions and virtually met with the study team at various times during training to provide updates on progress. If any issues regarding the compliance or ability of the participants to perform the exercise routines arose, the personal trainer(s) would contact the study team via call/text/email and a virtual meeting was set up for further discussion. Each week, exercise sessions consisted of a primary workout (session A) performed twice per week and a secondary workout (session B) performed once per week, separating the two primary sessions. The session A exercises consisted of the following: (1) goblet squat, (2) floor chest press, (3) Romanian deadlift, (4) bent-over row, (5) overhead triceps extension, (6) standing bicep curl, and (7) sit-ups. The session B exercises consisted of the following: (1) standing lunges, (2) shoulder press, (3) single-leg Romanian deadlift, (4) lateral raises, (5) bent-over triceps extension, (6) seated hammer curls, and (7) leg raises. The exercise program design and personal trainer oversight was handled by a member of the research team that was certified as a strength and conditioning specialist and exercise physiologist through the National Strength and Conditioning Association (NSCA) and the American College of Sports Medicine (ACSM), respectively.

The participants became familiarized with proper weightlifting technique prior to establishing a complete ten-repetition maximum (10RM) test for each exercise to prescribe loads. Due to the limitations of available equipment for home-based training, an emphasis was placed on volume progression instead of load progression, where set number was increased every 3–4 weeks unless individual performance dictated otherwise, in which case adjustments were made to best suit the participant. The set number progressed from 1–4 sets, with a repetition range of 8–15. The relative load (%1RM) remained constant at ~60% 1RM. In order to maintain a load of 60%, the weight was increased from subsequent sessions if the participant was able to complete ≥3 repetitions outside of the prescribed range (i.e., ≥18) during the last set of a particular exercise. Core training exercises (i.e., sit-ups and leg raises) did not use external weight and the prescribed repetition maximum was capped at 20 repetitions per set. The rest periods were 1–2 min between sets and exercises. The participants were asked to record the number of repetitions they completed for each set of each exercise in addition to the load lifted. The resistance training volume-loads (sets × repetitions × load) were totaled at the end of every session and summed up at the conclusion of every week.

### 2.5. Statistical Analysis

All data were analyzed using Microsoft Excel 2016 (Microsoft Corporation, Redmond, WA, USA) and SPSS for Windows (SPSS 26.0, Chicago, IL, USA). The compliance rates were derived from recorded logs kept by each personal trainer assigned to a participant for the duration of the RET program. The number of sessions completed by each participant in a 12-week period was divided by the total number of sessions assigned (i.e., 36). The home-based RET program was considered feasible based on the average compliance rate percentage of all participants. Paired samples *t*-test were conducted to determine significant changes in the body composition and physical fitness metrics from pre- to post-intervention. Cohen’s *d* statistic was used for effect sizes (ES), classified as 0.2, 0.5, and 0.8 for small, moderate, and large differences, respectively. All data were expressed as means ± standard deviations (M ± SD), unless otherwise indicated. Statistical significance was *p* < 0.05.

## 3. Results

### 3.1. Participant Characteristics

The baseline characteristics are presented in Table 1. A total of four boys and six girls (15.8 ± 2.2 yr, 164.0 ± 9.3 cm, 60.0 ± 15.1 kg) were enrolled and completed the study. The girls had higher BMI values at both pre- (25.1 ± 7.4 kg/m^2^) and post-intervention (25.4 ± 7.7 kg/m^2^) assessments than the boys (18.6 ± 1.8 and 19.0 ± 2.2 kg/m^2^, respectively), but no significant changes occurred during the intervention. Forced expiratory volume % for age significantly decreased from 109.1 ± 17.6% to 103.0 ± 16.3% (*n* = 8, ES = 0.36, *p* = 0.02), while a non-significant decrease was observed in forced vital capacity % for age (109.3 ± 14.3 to 104.8 ± 12.8%; ES = 0.33, *p* = 0.05).

### 3.2. Resistance Exercise Training

The participants’ compliance for completion of the RET can be seen in Table 2. Of the 36 resistance training sessions prescribed, 28.4 ± 3.8 sessions (78.9%) were completed. One participant experienced back pain resulting from the exercise training; therefore, post-testing took place at 8 weeks (19 sessions completed). After removing the outlier participant, compliance increased to 29.4 ± 2.1 sessions, or 81.8 ± 5.8%. Figure 1 depicts the volume-load achieved each week alongside the number of participants actively involved.

### 3.3. Metabolic Responses, Body Composition, and Physical Fitness Outcomes

Changes in the metabolic outcome variables can be seen in Table 3. Statistically significant increases were only observed with the 2-h C-peptide levels; however, a moderate decrease in fasting glucose levels and a moderate increase in 2-h insulin levels were also seen. Pre- to post-intervention in changes in the body composition metrics are displayed in Table 4. A statistically significant decrease occurred in %Fat, while significant increases were observed in FFM and FFMI; however, all changes were noted as having a small effect size. The results of the maximal isometric and isokinetic knee extension testing can be seen in Table 5. A small, but statistically significant increase was demonstrated in the average power during the 90°/s isokinetic knee extensions. The pre- to post-intervention results from the GXT can be seen in Table 6. Small, yet significant increases occurred in V̇O_2peak_, V̇CO_2peak_, and VE.

## 4. Discussion

The findings of the current study demonstrate the feasibility of a home-based RET program using a telehealth approach in adolescents with glucose intolerance and suggests positive effects on insulin secretion, body composition, and physical fitness in adolescents with CFRD. Compliance scores of 78.9% (all participants) and 81.8% (without outlier) were observed, with both results falling in line with previous research. A systematic review by Cox et al. [19] reviewed eight studies examining the efficacy of telehealth in people with CF. Non-compliance ranged from 43–63%, and adherence across four studies was reported at 52–80% [19]. In a recent study by Tomlinson et al. [20], nine adults with CF participated in a Skype-based exercise program for eight weeks. The mean compliance was 68%, with 25% of the sessions experiencing some level of technical difficulties [20]. The personal trainers utilized in the current study also reported instances of sessions either being delayed or completely missed due to technical issues with the software being used or internet connection errors. Additional observations from trainers were that longer periods of familiarization are necessary for adolescents with CF compared to their non-CF counterparts due to lower levels of fitness, the higher risk of an adverse event, and because the participants may be unfamiliar with the feeling of discomfort and the ability to properly exert oneself during exercise sessions. The best strategy for increasing compliance was noted as frequent encouragement for greater motivation and reinforcing the short-term and long-term goals of participation in the RET program.

Small effects were observed in glucose responses during OGTT testing. However, the most notable and significant change was the increase in 2-h C-peptide levels. This was observed in association with a moderate decrease in fasting glucose and an increase in 2-h insulin levels. Conversely, there was a small increase in 2-h glucose. Finally, no changes were observed in any of the insulin sensitivity markers (i.e., the Matsuda index, insulin/glucose iAUC, and OGIS). Previous research in non-CF participants has shown that RET is an effective strategy for enhancing insulin sensitivity. A study in adults with type 2 diabetes by Tavakol et al. [21] found that RET for 8 weeks led to improved glucose homeostasis through increased insulin secretion. The basis for RET relates to the ability of skeletal muscle contractile activity to enhance insulin sensitivity and promote glucose uptake and clearance. In the current study, significant reductions in percent body fat were seen alongside increases in fat-free mass and a small effect on lean tissue mass. These improvements in overall body composition may be the cause of the change in insulin secretion, demonstrating the importance of body composition and nutritional status in patients with CF, but further investigation is needed [22,23,24].

Mixed results were seen for metrics of pulmonary function and physical fitness in the current study. FEV1% and FVC1% both decreased following the intervention, but V̇O_2peak_, V̇CO_2peak_, and VE all significantly increased, indicating enhanced aerobic capacity; however, these increases produced small effects. It could be asserted that even though the observed increases were small, participation in RET can help to maintain or slightly raise aerobic capacity levels in the absence of traditional aerobic endurance exercise. Additionally, increases were observed in isokinetic strength measures, yet none were significant. Larger increases in strength measures may be due to the limitation of equipment utilized, which directly impacts the ability to apply progressive overload via the load lifted and the type and number of exercises implemented. A greater emphasis was placed on increasing set and rep volume week to week to counteract the limitation of available “heavy” loads. It is worth noting that completed volume-load (depicted in Figure 1) consistently increased week-to-week, giving the indication that, on average, the participants were able to perform a greater amount of “work” within the same 60-min training session. Previous studies have resulted in improvements in residual lung volume, forced expiratory volume, forced vital capacity, physical work capacity, and aerobic performance measured through V̇O_2peak_ [25,26,27,28,29,30]. It is suggested that the observed changes can be attributed to increased strength of the core muscles involved in respiratory function and increased endurance of the working muscles involved in the specific modality of exercise testing used to assess aerobic fitness. Although increases in aerobic capacity have been observed in those who are healthy, untrained or trained individuals, the most consistent improvements seem to occur in sedentary individuals with low baseline fitness levels (V̇O_2max_ ≤ 40 mL/kg/min).

The present study was conducted in a small sample size, limiting the statistical power. This limitation can greatly affect the resulting post-intervention changes, making it difficult to derive conclusions or provide recommendations with high confidence. Recruiting participants with CF can be difficult, especially when looking for individuals within a set age range. However, this study should be looked at as a pilot study, with the primary purpose being assessing the feasibility of utilizing telehealth to conduct an RET program with the targeted population. Moreover, the current results fall in line with previous studies and address an area of the literature with limited research. The novelty of this study can serve as a foundation for future research and practitioners seeking to implement RET using telehealth.

## 5. Conclusions

In conclusion, home-based RET using a telehealth approach is feasible in adolescents with CF. The current study suggests a positive impact on insulin secretion, body composition, and exercise capacity that will likely result in improved clinical outcomes in patients with CF and glucose intolerance. However, practitioners should anticipate the obstacles of technological difficulties, such as dropped calls, video freezing, and poor internet connection. Moreover, practitioners conducting exercise sessions will need to use individualized programming to fit the personal progression of the participant and be able to boost motivation through frequent and consistent encouragement. Though the physiological effects of RET are still not fully understood, the observed results may speak more to the specific protocol utilized than the efficacy of home-based RET programs as a whole. Future studies should experiment with the manipulation of exercise variables to examine how outcomes may change in this population.

## Figures and Tables

**Figure 1 ijerph-19-03297-f001:**
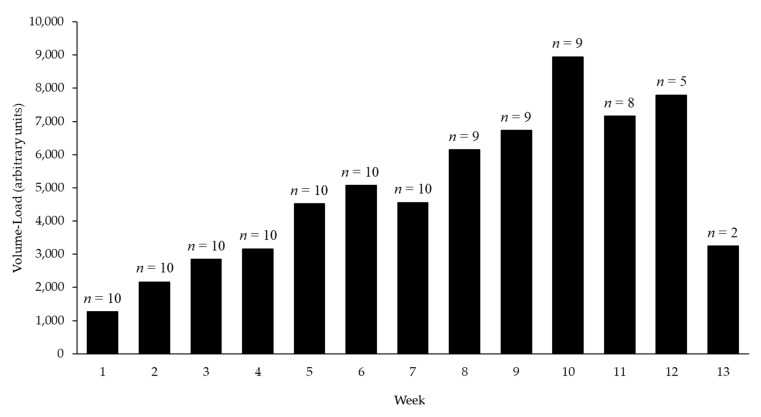
Average completed volume-load progression; *n* = number of participants who completed exercise sessions for that week.

**Table 1 ijerph-19-03297-t001:** Descriptive data.

Age (yr)	15.80 ± 2.20
Height (cm)	163.97 ± 9.31
Height Z-score	−0.21 ± 0.68
Weight (kg)	60.05 ± 15.12
Weight Z-score	−0.04 ± 1.58
Body Mass Index (kg/m^2^)	22.55 ± 6.52
Body Mass index Z-score	−0.09 ± 1.70
Fasting Glucose (mg/dL)	100.80 ± 11.30
2-h Glucose (mg/dL)	156.22 ± 45.41
Forced Expiratory Volume (FEV1%)	109.13 ± 17.64
Forced Vital Capacity (FVC1%)	109.25 ± 14.34

**Table 2 ijerph-19-03297-t002:** Resistance exercise session compliance.

Participant	Completed Sessions	Missed Sessions	Completed Percentage
001	31	5	86.1
002	30	6	83.3
003	30	6	83.3
004	28	8	77.8
005	29	7	80.6
006	32	4	88.9
007	31	5	86.1
008	19	17	52.8
009	29	7	80.6
010	25	11	69.4
M ± SD	28.4 ± 3.8	7.6 ± 3.8	78.9 ± 10.7

M = mean, SD = standard deviation.

**Table 3 ijerph-19-03297-t003:** Metabolic marker pre-post changes.

	*N*	Pre-Intervention	Post-Intervention	Mean Difference	ES	*p*-Value
Fasting Glucose (mg/dL)	10	100.8 ± 11.3	95.6 ± 5.3	−5.2	0.6	0.11
2-h Glucose (mg/dL)	9	156.2 ± 45.4	158.9 ± 40.4	2.7	0.1	0.84
Fasting Insulin (U/mL)	10	9.7 ± 5.5	11.3 ± 8.7	1.6	0.2	0.36
2-h Insulin (U/mL)	9	85.0 ± 42.6	120.0 ± 91.3	35.0	0.5	0.10
Fasting C-peptide (ng/mL)	10	2.0 ± 0.7	2.1 ± 1.2	0.1	0.2	0.54
2-h C-peptide (ng/mL)	9	9.6 ± 3.0	11.7 ± 4.9	2.1 *	0.5	0.04
Matsuda Index	9	4.1 ± 2.5	4.0 ± 3.7	−0.1	0.03	0.93
HOMA-IR	9	2.4 ± 1.3	2.8 ± 2.1	0.4	0.3	0.35
OGIS	8	363.6 ± 42.0	362.4 ± 75.9	−1.2	0.02	0.96
Insulin/Glucose-iAUC	9	0.6 ± 0.5	0.6 ± 0.4	−0.1	0.2	0.59

HOMA-IR = homeostatic model assessment of insulin resistance; OGIS = oral glucose insulin sensitivity; iAUC = incremental area under the curve; * significant alpha level at *p* < 0.05.

**Table 4 ijerph-19-03297-t004:** Body composition pre-post changes.

	*N*	Pre-Intervention	Post-Intervention	Mean Difference	ES	*p*-Value
%Fat	10	29.7 ± 14.7	28.4 ± 15.3	−1.3 *	0.1	0.03
FM	10	19.3 ± 14.1	19.0 ± 15.0	−0.3	0.02	0.60
FFM	10	40.8 ± 7.2	42.3 ± 7.2	1.5 *	0.2	0.01
LBM	10	39.6 ± 7.1	40.6 ± 7.0	1.0	0.1	0.09
FMI	10	6.8 ± 5.5	6.0 ± 6.2	−0.8	0.1	0.23
FFMI	10	14.1 ± 1.8	14.5 ± 1.9	0.4 *	0.2	0.01
LBMI	10	13.4 ± 1.8	12.5 ± 5.0	−0.9	0.3	0.52
FMI-Z-score	10	−0.7 ± 1.6	−0.9 ± 1.9	−0.2	0.1	0.13
LBMI-Z-score	10	−0.8 ± 1.2	−0.9 ± 1.2	−0.1	0.04	0.88

%Fat = percent body fat; FFM = fat-free mass; FFMI = fat-free mass index; FM = fat mass; FMI = fat mass index; LBM = lean body mass; LBMI = lean body mass index; * significant alpha level at *p* < 0.05.

**Table 5 ijerph-19-03297-t005:** Isometric and isokinetic knee extension test metric pre-post changes (*N* = 10).

Isometric	Pre-Intervention	Post-Intervention	Mean Difference	ES	*p*-Value
Peak Torque (N·m)	117.7 ± 37.8	124.8 ± 32.1	7.1 ± 13.3	0.2	0.13
Peak Torque (N·m/kg)	2.0 ± 0.7	2.1 ± 0.7	0.1 ± 0.2	0.1	0.27
Time to Peak Torque (s)	3.0 ± 1.0	2.8 ± 1.0	−0.3 ± 0.9	0.3	0.40
**Isokinetic 90°/s**					
Peak Torque (N·m)	9.9 ± 35.1	95.3 ± 41.9	5.4 ± 24.0	0.1	0.50
Peak Torque (N·m/kg)	1.6 ± 0.6	1.6 ± 0.8	0.1 ± 0.5	0.1	0.64
Work Done (J)	85.7 ± 42.5	98.4 ± 41.7	12.7 ± 23.1	0.3	0.12
Average Power (W)	73.4 ± 38.6	89.5 ± 39.8	16.2 ± 17.7 *	0.4	0.02
Time to Peak Torque (s)	0.4 ± 0.2	0.4 ± 0.2	0.01 ± 0.2	0.2	0.91
**Isokinetic 180°/s**					
Peak Torque (N·m)	70.4 ± 39.9	71.8 ± 32.9	1.3 ± 25.0	0.04	0.87
Peak Torque (N·m/kg)	1.2 ± 0.7	1.2 ± 0.6	−0.01 ± 0.5	0.02	0.94
Work Done (J)	67.1 ± 46.2	76.2 ± 39.1	9.9 ± 24.3	0.2	0.23
Average Power (W)	98.0 ± 74.6	113.6 ± 63.6	21.0 ± 47.0	0.2	0.19
Time to Peak Torque (s)	0.3 ± 0.1	0.3 ± 0.1	−0.04 ± 0.2	0.00	0.54
**Isokinetic 270°/s**					
Peak Torque (N·m)	62.1 ± 38.9	63.4 ± 29.5	1.6 ± 17.5	0.04	0.78
Peak Torque (N·m/kg)	1.1 ± 0.70	1.1 ± 0.6	−0.02 ± 0.3	0.03	0.88
Work Done (J)	52.0 ± 38.6	61.9 ± 35.7	6.9 ± 15.0	0.3	0.18
Average Power (W)	103.2 ± 86.2	124.1 ± 75.6	18.2 ± 43.7	0.4	0.22
Time to Peak Torque (s)	0.3 ± 0.2	0.2 ± 0.1	−0.03 ± 0.2	0.3	0.56
**Isokinetic 360°/s**					
Peak Torque (N·m)	58.9 ± 31.1	60.5 ± 24.6	1.6 ± 17.5	0.1	0.78
Peak Torque (N·m/kg)	1.0 ± 0.6	1.0 ± 0.4	−0.02 ± 0.3	0.02	0.88
Work Done (J)	46.0 ± 33.0	52.9 ± 27.4	6.9 ± 15.0	0.2	0.18
Average Power (W)	106.0 ± 100.0	124.2 ± 74.6	18.2 ± 43.7	0.2	0.22
Time to Peak Torque (s)	0.3 ± 0.2	0.3 ± 0.1	−0.03 ± 0.2	0.2	0.56

* significant alpha level at *p* < 0.05.

**Table 6 ijerph-19-03297-t006:** Cycle ergometer V̇O_2peak_ test metric pre-post changes.

	*N*	Pre-Intervention	Post-Intervention	Mean Difference	ES	*p*-Value
V̇O_2peak_ (L/min)	10	1.7 ± 0.5	1.9 ± 0.6	0.1 ± 0.1 *	0.2	0.01
V̇O_2peak_ (mL/kg/min)	10	30.0 ± 10.0	31.7 ± 11.2	1.7 ± 3.1	0.2	0.11
V̇CO_2peak_ (L/min)	10	2.1 ± 0.7	2.2 ± 0.7	0.1 ± 0.1 *	0.2	0.01
V̇CO_2peak_ (mL/kg/min)	10	35.9 ± 12.8	37.1 ± 13.6	1.2 ± 2.0	0.1	0.11
RER	10	1.2 ± 0.1	1.2 ± 0.1	−0.03 ± 0.1	0.3	0.29
V̇E (L/min)	10	51.8 ± 11.3	57.1 ± 16.8	5.3 ± 6.8 *	0.4	0.04
MWR (W)	10	137.0 ± 49.0	135.0 ± 47.9	−2.0 ± 20.4	0.04	0.76
MWR (W/kg)	10	2.4 ± 0.9	2.3 ± 0.9	−0.1 ± 0.4	0.1	0.59
HR_max_ (bpm)	10	179.5 ± 12.8	180.5 ± 16.9	1.0 ± 5.3	0.1	0.57
Age-Predicted HR_max_ (%)	10	87.9 ± 5.9	88.5 ± 8.0	0.6 ± 2.7	0.1	0.51
RPE	9	18.9 ± 1.3	17.7 ± 1.3	−1.4 ± 2.0	0.9	0.06
HRR @ 5-min (bpm)	8	121.3 ± 13.9	119.5 ± 12.7	−2.9 ± 6.7	0.1	0.26

HR_max_ = maximum heart rate; MWR = maximal work rate; RER = respiratory exchange ratio; RPE = ratings of perceived exertion; VE = ventilation; V̇CO_2peak_ = peak carbon dioxide expiration; V̇O_2peak_ = peak oxygen consumption; * significant alpha level at *p* < 0.05.

## Data Availability

This study did not report any public data.

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
