# Peer review of "Feasibility and Efficacy of Telehealth-Based Resistance Exercise Training in Adolescents with Cystic Fibrosis and Glucose Intolerance"

_ijerph, 2022, doi:10.3390/ijerph19063297_

Round 1

Reviewer 1 Report

I congratulate Holmes et al. for presenting such a nice study highlighting the importance of telemedicine for the management of cystic fibrosis. The study is very relevant in the current COVID-19 era. 

However, I have some concerns regarding the data, especially the glucose function.

1) It seems that the 12-14 weeks of intervention did not have any change in the insulin secretion rate as the C peptide levels did not change at both fasting and at 2 hours but the author conclude this in the abstract as well as in the discussion. Can the authors explain what method they are using for this conclusion? The 2 hr insulin increase can be attributed to reduced insulin sensitivity. The authors can confirm this using indirect measures such as HOMA IR and QUICKI etc. Since you only need fasting glucose and insulin values to calculate these indirect insulin sensitivity measures. 

2) I feel it strange that the authors have not reported the changes in the body composition (FFM, FFMI, LBM etc)  in a table form or as a figure. These are one of your main results I feel that by not highlighting them properly you are making your paper weak. 

3) Even though participants were not diabetic I was wondering if the authors has taken the long term blood sugar levels at baseline (HbA1c). Adding this would significantly highlight the impact of telehealth on glucose metabolism

My minor concerns 

1) line 21 please add after A compliance score of 78.9% (all participants) 

2) line 207 the authors have written twice FFM in the same line. I assume one of is FFMI. Please correct this

3)  May be I missed it but I could not find what was included in Session A and Session B in the material and methods section.

Author Response

Please see the following major changes:

Abstract:

Lines 21-28

Methods:

Lines 148-171

Results: 

Tables 1, 3, and 4

Lines 228-238

Discussion

Lines 277-282

Lines 295-298

Lines 310-319

Reviewer 2 Report

  1. General comments

The purpose of this study was twofold: 1) determine the feasibility of a home-based resistance 15 exercise training program in patients with cystic fibrosis and impaired glucose tolerance using virtual personal training, and 2) observe the effects completion of the RET program had on glucose metabolism, pulmonary function, body composition, and physical fitness. This prospective study is based on a small sample of adolescents with cystic fibrosis and shed light on the feasibility and efficacy of live online resistance training. The manuscript is very well written, but authors should further describe the online exercise program.

  1. Specific comments

2.1. Abstract

Please, indicate the p-value of each result.

2.2. Introduction

Line 42: Please, rephrase the sentence. I missed when I read “established method for is an effective strategy”.

2.3. Materials and Methods

Line 75: Please, write the indication to Table 1 in the results section.

Line 95: Please, put the Table in the results section. Furthermore, indicate how many participants were underweight, healthy weight, overweight or obese.

Line 134: The authors should considered to detail the intervention following the consensus on exercise reporting template (doi: 10.1136/bjsports-2016-096651) in order to allow a better replication by other researchers.

Please, indicate the way that authors communicate with the participants to join the online personal training sessions (e.g., social media, web platform, mobile app).

Please, indicate how the personal trainers delivered the exercise sessions (i.e., in single or group-based setting).

Please, indicate whether (or not) personal trainers gave any kind of feedback to participants.

Line 144: Please, correct the word ‘overheard’ by ‘overhead’.

2.4. Results

Line 207: The authors have written the same word (i.e., FFM) twice. Please, write the sentence well.

Please, include a table (or figure) with the results on metabolic response, body composition and physical fitness.

Line 209: Please, describe the results shown in Table 3 and 4.

2.5. Discussion

Line 228: Please, indicate the software you used in method section.

Line 233: Please, indicate this information in method section.

Line 275: Please, include the strength and limitation section. Perhaps, sample size and the non-controlled design should be stated.

Author Response

(The authors gave the same response as above.)

Reviewer 3 Report

In this manuscript authors determined the feasibility of a home-based resistance exercise training (RET) program in adolescents with cystic fibrosis (CF) and evaluated the effects of the RET program on several clinically relevant outcomes. The study is interesting and could provide some useful information. However, the strength of the study is limited by several weaknesses.

This is a small non-randomized and non-controlled study and thus does not fill the gap outlined in lines 48-51. The lack of a control group and of a sample size estimation limits the impact of the study. Perhaps, in the title should be specified that this is “a pilot study”.

The authors should better justified the use of a RET program instead of an aerobic endurance training program since two of the objectives of the study were to increase maximal oxygen consumption and to enhance glycemic control.

The lack of effects of RET on isometric and isokinetic strength is rather surprising. The authors are invited to go deeper on this issue.

Material and Methods: the number of the patients evaluted should be included in this section. In the Results section FVC and FEV data are presented but no mention that patients performed pulmonary function tests is done.

You claim the RET program induced a significant incease of VO2peak. However, the absolute increase (0.13 L/min on average) could be smaller than the technical error of measurement. Please, report the TEM for VO2 in your Lab. In any case, despite the increase in VO2peak, patients were able to perform the same peak work rate. In my opinion this finding does not suggest an enhanced aerobic capacity (Line 250). Please deeply discuss this finding  

Minor comments

Table 1: the number of decimals for some variables should be limited to one

Table 4: Is rather unusual to normalize CO2 output for kilogram of body mass

Lines 125-136: “10 W/min”: please delete “/min”

Line 129: Please, insert “Peak” oxygen uptake values

Line 148: “a complete a ten-repetition maximum (10RM) test”: please delete “a”’.

Author Response

(The authors gave the same response as above.)

Round 2

Reviewer 1 Report

I am glad that you added the C peptide values. This makes your results even more important. 

Can you please re-edit lines 314-315 and rephrase it by just saying no changes were observed in any of the insulin sensitivity parameters. Rather than phrasing it as you have done now.

Author Response

Reviewer #1

I am glad that you added the C peptide values. This makes your results even more important. 

-Thank you for the suggestion!

Can you please re-edit lines 314-315 and rephrase it by just saying no changes were observed in any of the insulin sensitivity parameters. Rather than phrasing it as you have done now.

-Revised.

Reviewer 2 Report

Authors have made all the changes suggested.

I would accept the manuscript in the present form.

Author Response

Reviewer #2

Authors have made all the changes suggested. I would accept the manuscript in the present form.

-We thank you for your review and acceptance of our work!

Reviewer 3 Report

regretfully, the revised version of the manuscript in object has not achieved an acceptable quality.
In the cover letter the authors did not address the raised concerns but simply listed the changes made in the manuscript

Author Response

Reviewer #3

Regretfully, the revised version of the manuscript in object has not achieved an acceptable quality.
In the cover letter the authors did not address the raised concerns but simply listed the changes made in the manuscript

-The authors would like to apologize to the reviewer for not addressing their comments. Please see below and we thank you for taking the time to review our work. However, we respectfully disagree on the quality of our work and believe that the previous changes and the additional ones made address all major points of concern.

This is a small non-randomized and non-controlled study and thus does not fill the gap outlined in lines 48-51. The lack of a control group and of a sample size estimation limits the impact of the study. Perhaps, in the title should be specified that this is “a pilot study”.

-The word “randomized” has been removed in Line 56. Barring this one word, the current study does address a gap in the literature.

-The word “pilot” has been inserted into Line 72. This in addition to the “feasibility aspect of the purpose statement and Lines 309-318 should indicate that more research is needed, but does not discount the current work.

The authors should better justified the use of a RET program instead of an aerobic endurance training program since two of the objectives of the study were to increase maximal oxygen consumption and to enhance glycemic control.

-Referring back to the Introduction, the resistance exercise has not been deeply explored in this population even though this modality of exercise is widely utilized for many benefits, including favorable changes in body composition and muscular fitness. There is existing evidence of increased glucose control via resistance training.

The lack of effects of RET on isometric and isokinetic strength is rather surprising. The authors are invited to go deeper on this issue.

-See Lines 297-305. Also, the exercise program implements may not have transferred over to the maximal leg extensions done on the Biodex used for testing. Many studies using resistance exercise, utilize machine based movements like “Leg Extensions” which directly translate to post-intervention assessments.

Material and Methods: the number of the patients evaluted should be included in this section.

-This is a preference not a rule. The authors respectfully decline and point to the number listed in the Results.

In the Results section FVC and FEV data are presented but no mention that patients performed pulmonary function tests is done.

-Added.

You claim the RET program induced a significant incease of VO2peak. However, the absolute increase (0.13 L/min on average) could be smaller than the technical error of measurement. Please, report the TEM for VO2 in your Lab. In any case, despite the increase in VO2peak, patients were able to perform the same peak work rate. In my opinion this finding does not suggest an enhanced aerobic capacity (Line 250). Please deeply discuss this finding  

-This information is not available. However, see Lines 294-297. The fact that aerobic capacity and work rate, at bare minimum, can be maintained following 14 weeks of resistance training is interesting information and becomes even more valuable when looking at a population of CF patients.